# Modeling the Effects of Seed Maturity on Cooking Time of ‘Dimitra’ Lentils

**DOI:** 10.3390/foods12010042

**Published:** 2022-12-22

**Authors:** Maria Svarna, Athanasios Mavromatis, Dimitrios N. Vlachostergios, Dimitrios Gerasopoulos

**Affiliations:** 1Laboratory of Food Processing and Engineering, Department of Food Science and Technology, Faculty of Agriculture, Aristotle University of Thessaloniki, 54124 Thessaloniki, Greece; 2Laboratory of Genetics & Plant Breeding, Faculty of Agriculture, Aristotle University of Thessaloniki, 54124 Thessaloniki, Greece; 3Institute of Industrial and Forage Crops, Hellenic Agricultural Organization Demeter, 41335 Larissa, Greece

**Keywords:** cooking time, seed maturity, optimal cooking time, *Lens culinaris* Medik, texture analysis, organoleptic method, absolute positive force

## Abstract

The lentil is a valuable crop for human nutrition and is cooked to adequate softening prior to consumption. The objective of the study was to use a model to point out the effects of seed maturity on optimum cooking time (OCT). Two lentil seed samples (cv ‘Dimitra’) exhibiting short (SCT) and long (LCT) cooking times (CT) were visually separated into brown- and green-colored categories, corresponding to mature and immature seeds, respectively. The 1000-seed mass and the percentages of maturity categories were measured in samples before they were subjected to 20–60 min CT. Absolute positive force (APF)-based texture analysis parameters were monitored during CT. OCT thresholds were established by correlating the organoleptic with the texture analyzer parameters. The averaging and weighted averaging of the texture analysis parameters, or even their modeling, failed to produce a realistic OCT due to texture values exceeding the OCT threshold. However, the modeling of the percentage of cooked seeds during CT predicted a realistic OCT, which was also validated later. In this model, all seeds (overcooked or intact, mature or immature) were taken into account. Among the texture parameters, APF better described cooking. Mature seeds softened faster and produced more overcooked seeds than did the immature seeds. The different proportions of maturity categories within the SCT and LCT seeds greatly affected the sample OCT.

## 1. Introduction

The lentil (*Lens culinaris* Medik.) is a very important crop for human nutrition due to its rich composition of proteins, carbohydrates (dietary fiber), minerals, and vitamins, including a wide range of bioactive phytochemicals [1,2]. Lentils are usually consumed in a cooked form [3]; therefore, seed softness, among other sensory quality traits, has been recognized as the most important commercial trait [4,5,6]. Cooking quality or cooking doneness is predominantly defined as the boiling or cooking time (CT) required to render the seeds adequately soft for consumption. Softening occurs following water penetration into the cotyledon through the seed coat during cooking. The shorter the CT, the better the cooking quality and the lower the energy and time required [7].

Several factors affect cooking quality. It is well known that the genetic factor affects the seed chemical composition and physicochemical characteristics, in addition to seed size and seed processing, which includes soaking, cooking, or even dehulling [8]. Further, the genetic factor has been reported to control seed coat anatomy and the ability of hot water to penetrate the cotyledon. This in turn causes the variability of CT in lentil varieties [9,10]. Apart from genotypic factors, the preharvest soil-climatic or cultural (such as location and growth season), in addition to the postharvest factors (such as storage) also affect CT [6,11,12,13]. Further, Theologidou et al. [6] reported significant correlations between the energy required to soften lentil seeds and the seed traits in a green-seeded lentil cultivar, possibly indicating a differentiation between mature and immature seeds in respect to CT. Lentil seeds are usually composed of a varying number of seed categories based on their harvest maturity, as determined by seed color; lentil seeds range from brown to green, considered to correspond to mature to immature seeds, respectively. These two (mature and immature), as well as more intermediate categories, are produced from lentil plants due to their habit of flowering in waves. The initial flowering waves possess adequate time and resampling to produce mature seeds, while the later waves may not, due to the changing of the environmental or other (cultivation or soil) conditions. Different lentil cultivars respond differently to these conditions and therefore, the seeds of these later waves may remain immature, to a certain extent. Upon harvest, the seeds could be considered as a mixture of the mature to immature seed categories. To our knowledge there are virtually no reports regarding the CT of lentil seeds of different maturity or CT in which seed maturities are taken into consideration.

Based on the above factors, the CT shows great variation among cultivars or among areas of cultivation for the same cultivar; Vlachostergios et al. [13] reported that the CT of Greek varieties ranged between 22.5–30.5 min, while in Turkish lentil varieties, the CT was recorded between 15.2 and 23.9 min [14]; Jood et al. [15] reported CT values ranging between 38 and 43 min, while Erkine et al. [16] reported an average CT of 33 min.

Another example of the reported variation in CT is related to the CT method itself. Most of the methods used are adapted from methodology developed for pulses, mainly beans. An extensive report on such methodology has recently been reviewed by Wood [16]. Even though lentil seeds, among many pulses, share common characteristics with beans, its peculiar size and shape require special attention in relation to CT determination. Among different methods applied in research works, Iliadis [7,12] has described a method by which in a limited number of 10 seeds per CT were subjected to needle intrusion. The needle course within the seed (depth) was measured using a penetrometer upon a loading of 50 g, at gravity of 0.2 s. Seeds were considered as cooked when the penetration value was 4 mm. This method has also been used by Ninou et al. [8]. In another method [6], seeds (20 in number) were cooked for 27 min and subjected to a puncture test using a texture analyzer equipped with multiple chip rigs. The seeds were then penetrated by the array of 2 mm diam cylindrical probes, at crosshead speed of 0.8 mm s^−1^. Cooking quality was estimated from the area under the force vs. time displacement curve (puncture energy) and expressed in mJ g^−1^. Scanlon et al. [17] used a texture analyzer equipped with a 1000 N load cell, a 10 cm^2^ wire extrusion cell, and an 8 bar extrusion grid operating at a crosshead speed of 60 mm.min^−1^. In this method, a cooked lentil sample of 10 g was subjected to compression testing at 60% of its original height using a 10 cm plunger, and the peak force was used as the CT. Compression of 40% is known to simulate initial molar bite [18]. A texture analyzer equipped with a Kramer shear cell holder has also been employed in a method used by Wang et al. [19]. According to this method, 75 g of cooked lentils, following cooling to ambient temperatures, were loaded onto a Kramer shear cell holder operating at 1.5 mm/s arm speed during compression. Firmness, as related to CT, was defined as the maximum shear force and expressed as N.g^−1^. In another study, Wang and Daun [20] used an automated Mattson bean cooker apparatus using individual seeds during cooking. The recorded time required for each plunger to drop was used to determine CT (when 80% of the seeds had been penetrated). The results of the Mattson cooker method agreed with those of a tactile method for determining cooking times. Vlachostergios et al. [13] used the tactile method to determine the CT of lentils (5 g) during cooking, sampling for seed softness every 30 s [21]. As the optimal cooking time (OCT) was considered as the time at which the seeds were soft enough to make a uniform transparent mass (with no opaque core in the seeds) when placed between two glass slides, pressed against each other with no lateral movement. Scanlon et al. [17] used another sensory method, that of chewiness. In this method the optimal CT was considered as the time at which the seeds were soft enough to easily yield to the first bite of molar pressure, subsequently breaking down into small particles during continued mastication.

The hard-to-cook phenomenon is common in grain legumes and has also been reported for lentils [22]. Bhatty [23] reported that in field-grown lentils, the hard-to-cook phenomenon may stem solely from cotyledons and not the seed coat, which can only reduce water uptake during cooking. Galiotou-Panayotou et al. [24] proposed a theory according to which hard-to-cook legume seeds are thought to develop based on the phytase–phytate–pectin interaction.

The aim of this study was to investigate the effect of seed maturity regarding two seed samples differing in OCT. Further, to overcome variation in determining the OCT of lentil seeds, the study aimed to propose a new model-based approach using objective and subjective tools.

## 2. Materials and Methods

### 2.1. Plant Materials

Seed samples of a commercial cultivar cv Dimitra (*Lens culinaris* Medik.) were used. ‘Dimitra’ is a medium-early cultivar with a rather small seed size and a yellow cotyledon color. The seed samples were produced at Domokos and Patriki (Greece) in the 2019/2020 growing season. The characteristics of both locations, along with yield and seed quality traits, are described by Vlachostergios et al. [13]. Seeds produced at the Domokos area were characterized by a short CT (SCT), while seeds from Patriki were characterized by a long CT (LCT). The harvested lentil seeds contained 13% moisture and were stored at room conditions (21 °C and 50% relative humidity) for 5 months before the cooking quality study began.

### 2.2. Agronomic and Morphological Seed Characteristics

The 1000-seed mass was determined by counting 5 × 200 intact lentil seeds using a scale. Seed moisture content was calculated by comparing fresh and dry weight.

Intact lentil seed color was measured by a Minolta CR-410 chroma meter (Minolta, Osaka, Japan) equipped with an 8 mm measuring head and a C illuminant (6774 K). The colorimeter was calibrated using the manufacturer’s standard white plate. Quantified color change data were collected for L*, a*, and b* color space. L* refers to lightness, ranging from 0 = black to 100 = white; a* = redness/greenness, b* = yellowness/blueness. All color parameters for each sample were the instrument average of six independent measurements.

### 2.3. Assessment of Cooking Quality

#### 2.3.1. Cooking Procedure

For each seed sample (SCT and LCT), 10 g-subsamples were counted and divided into two categories based on their seed color as a maturity indicator: brown and green seeds, called hereafter MB (MB) and IMG (IMG) seeds, respectively. The seeds of each maturity category were counted, weighed, and placed separately in nets. Each subsample (the nets with MB and IMG seed categories) was then rinsed in deionized water and placed in a 250 mL glass Pyrex bottle with a screw cap, and 150 mL of boiling distilled water was added. The cap was placed on the bottle, but was not closed. The bottle was immersed in a covered boiling water bath (100 °C), heated on a hot plate, and cooked for 20, 30, 40, 50, and 60 min (CT). A different subsample was used for each CT. After cooking, the lentils were drained and cooled to room temperature on a paper towel. Then the seeds, per the MB and IMG categories, were weighed and the overcooked seeds, defined as seeds with a torn seed coat and a split or mashed cotyledon, were counted.

#### 2.3.2. Texture Analysis

Cooked seeds, cooled down to room temperature, were placed on the base of a single chip rig (TA.XT plus C, Stable Micro Systems). For each sample, 15 individual typical seeds per MB and IMG for the ‘Dimitra’ cultivar category were used. Each seed was penetrated by a thin cylindrical needle probe of 2 mm diam at a crosshead speed of 0.8 mm.s^−1^. A sample of 5–10 overcooked seeds was measured separately. The texture analysis parameters were expressed as absolute positive force (APF) required to penetrate the seed (N), the area to absolute positive force (AAPF), indicating the work required to penetrate the seed (N.s), and the gradient to APF (GAPF) (N.s^−1^).

#### 2.3.3. Establishing Optimal Cooking Time (OCT)

The samples of lentil seed originated from two different areas, divided into 3 subsample replications of 10 g, were cooked for an adjusted time of 27 min [6], selected as the cooking time for the screening. After cooking 10 individual seeds per replication, they were subjected to texture analysis, as described above.

Organoleptic texture (tactile and chewiness) was evaluated by a five-member panel (graduate students and staff). Tactile texture was evaluated in 10 samples per replication by pressing at least three seeds, one at a time, between the thumb and forefinger [13]. Chewiness was evaluated in 10 samples per replication by chewing three seeds per sample using the molars, breaking down the sample into small particles during mastication. Both methods were rated according to 1–9 scale:

1 = uncooked—seed is very difficult or not smashable, and cotyledon feels very hard;

2 = moderately undercooked—seed is difficult to smash, and cotyledon feels hard enough;

3 = slightly undercooked—seed is less difficult to smash, and cotyledon feels moderately hard;

4 = very slightly undercooked—seed is a little difficult to smash and cotyledon feels a little hard;

5 = rather cooked—seed is firm, but smashes relatively easily and cotyledon feels neither soft nor hard;

6 = very slightly overcooked—there is some resistance to smashing, and cotyledon feels a little mushy;

7 = slightly overcooked—there is little resistance to smashing and cotyledon feels nearly mushy;

8 = moderately overcooked—there is very little resistance to smashing, and cotyledon feels mushy;

9 = overcooked—seed is already mushy or overcooked, or presses into a mush upon application of pressure.

Organoleptic texture (tactile and chewiness) data was correlated with the texture analyzer readings following a linear model. Optimal CT (OCT) threshold values were produced for each texture analysis parameter (APF, AAPF, and GAPF).

### 2.4. Model Development

In a first approach, to predict the OCT, texture analyzer data (as the average and weighted average) were plotted against the CT using an exponential decay model, for both SCT and LCT seed samples. OCT was obtained by the use of the OCT threshold values for both seed maturity categories (MB and IMG), or for the total seed, as well as for both SCT and LCT seed samples.

In a second approach, the distribution of texture analyzer readings was used to calculate the percentage of cooked seeds from the texture analysis values that were within the range of zero to the OCT threshold value for each texture analysis parameter (APF, AAPF, and GAPF). These data were fitted in a sigmoid model from which the OCT could be predicted.

### 2.5. Validation for OCT Prediction Model

Both seed samples were cooked in a different set of samples at predicted OCTs according to the model of the second approach. The seeds were then subjected to organoleptic and texture analysis to validate the OCT predicted by the model.

### 2.6. Experimental Design and Statistical Analyses

For this experiment a complete randomized factorial design (2 × 2 × 5) was used, with three replicates per treatment. The factors were the seed sample (SCT and LCT), seed maturity (MB and IMG), and cooking time (20, 30, 40, 50, and 60 min).

Data were subjected to analysis of variance (ANOVA) and modeling using the statistical program SPSS v.25 and Microsoft Excel. The means were separated with Duncan’s new multiple range test (*p* ≤ 0.05). The effect size of each factor was evaluated using η^2^ (eta squared) calculated as follows: η^2^ = SS factor/SS total, where SS = sum of squares.

## 3. Results and Discussion

This study intends to draw researcher ‘s attention to the flowering and seed development of lentil plants; according to this study, the early flowering waves allow for the production of mature seeds of the typical brownish color. On the other hand, the later waves often produce immature seeds (of greenish color tones), depending on the interaction of genetic, cultural, and the changing seasonal environmental conditions [25,26]. However, at harvest, all seed maturities are pooled together as a unified sample. In this study, two seed samples of the ‘Dimitra’ cultivar were selected based on a previous evaluation of their OCT—short CT (SCT) and long CT (LCT). From each seed sample, two maturity categories of seeds were separated, based on their color: The mature seeds of brownish color (MB) and immature ones of greenish color (IMG).

In Table 1, an initial characterization of SCT and LCT seed samples under study is presented: the mass of 1000 seeds was 24.4 and 34.1 g, for SCT and LCT, respectively. This indicated that SCT seeds contained a greater number of seeds at a given mass and therefore, a smaller size compared to the LCT seeds. Bhatty, [27] has reported that the cooking of lentil seeds is affected by the seed size, among other factors. The mass of 1000 seeds for the MB seeds for both the SCT and LCT samples was about 33 g. However, the mass of 1000 seeds for the IMG seeds was as low as 16.6 g and as high as 35 g for the SCT and the LCT, respectively. The mass of 1000 seeds has been reported to range from 20 to 40 g among different lentil seed varieties [8,13]. The lentil seed samples consisted of MB and IMG seeds at a ratio of 0.9 and 3 for the SCT and LCT seed samples, respectively (Table 1). The color of each seed sample was related to redness (a*) and lightness values, but not to yellowness; LCT had lower a* and higher L* values, which indicated more greenish tones for the LCT seeds (Table 1). The values of color attributes are also within the range of the Greek varieties reported by Ninou et al. [8]. All differences between the two seed maturities examined also reflected in the differences between the two seed samples (SCT and LCT).

### 3.1. Increase in Seed Mass and Overcooked Seeds during Cooking

Upon seed cooking, both SCT and LCT seed samples exhibited an increase in their mass due to water absorption. It is known that legumes absorb water during cooking [28]. Most of this increase took place within the first 20 min, as the seeds doubled their mass (Figure 1A). The seeds did not show any water absorption after the 40th minute of cooking. Wang et al. [29] also reported a higher water absorption at the early cooking stages in dry beans. MB seeds of both the SCT and LCT seed samples showed higher water absorption than the IMG seeds. Further, the SCT seeds showed faster and higher water absorption than that of the LCT (Figure 1A).

A similar pattern was observed in the percentage of the increase in the number of overcooked seeds (Figure 1B). Overcooked seeds included seeds with a torn seed coat, and a split and mushy cotyledon. The increase in seed mass by water uptake, as well as in the percentage of overcooked seeds, was also reported previously by Varoquaux et al. [30]. Again, the MB seeds showed a higher percentage of overcooked seeds than did the IMG in both seed samples. Further, the increase in the percentage of SCT-overcooked seeds (MB, IMG or the weighted sum of both) was higher than that of the LCT (Figure 1B).

### 3.2. Seed Texture Analysis during Cooking

A set of 15 individual seeds per seed sample and maturity category was subjected to texture analysis at each CT to determine the parameters: absolute positive force (APF), area to absolute positive force (AAPF), and gradient to absolute positive force (GAPF). The analysis of variance indicated that all texture analysis parameters (Table 2) were predominantly influenced by the CT factor, since this factor accounted most of the variation (η^2^: 58.55–63.18). The seed sample factor also had a significant influence on all three texture parameters, while the seed maturity factor significantly influenced (η^2^: 25–27) only the APF and AAPF (*p* > 0.001) parameters. The only significant interaction of the factors observed was exhibited by the seed sample x CT interaction for all seed texture parameters at η^2^ range of 4.5–8.22. Between the two seed samples, the SCT seeds exhibited significantly lower values than did the LCT in all three texture analysis parameters. As expected, the highest values for all texture parameters were observed at the shortest CT (20 min), while the lowest values were obtained following cooking for 50–60 min. Regarding the seed maturity, the MB seeds were different than the IMG seeds for the APF and AAPF, but not for the AAPF parameters (Table 2). The differences within each texture parameter derived from the interaction (seed sample × seed maturity × CT) are shown in Table 3.

### 3.3. Organoleptic Test

To establish the optimum CT (OCT), an organoleptic value of 6.5 (scale 1–9) was adopted for the adequate cooking doneness of the lentil seeds. Values of both tactile texture and chewiness were correlated with the texture parameters of APF, AAPF, and GAPF, and the results are presented in Figure 2. The corresponding range of texture parameter threshold values to the adopted organoleptic value of 6.5 (very slightly to slightly overcooked, with a little resistance to smashing, and the cotyledon feels a little to nearly mushy, but tender) was based on a 95% confidence interval, as indicated by the respective arrows in Figure 2: for APF, this was found to be ~0.85–0.90 N and 0.70–0.80 N for chewiness and tactile texture, respectively (Figure 2A,B). Thus, the value of 0.8 N APF, being an intermediate value between the two organoleptic tests, was used in further data processing as an APF threshold for the determination of CT. Similarly, the value of ~4.45 N.s was used for AAPF (Figure 2C,D), and ~0.1 N.s^−1^ was used for GAPF (Figure 2E,F).

### 3.4. Modeling of Average Values of Texture Analysis Parameters during CT

Table 3 shows the average APF, AAPF, and GAPF readings obtained following penetration of a 2 mm needle in individual lentil seeds, previously subjected to cooking at five different cooking times (20–60 min).

The average APF of SCT values of MB seeds (15 readings of intact plus 1 of overcooked seeds) decreased from 0.76 at 20 min to 0.38 N at 60 min, while the average APF of IMG seeds (15 readings of intact seed plus 2–3 of overcooked seeds, depending on initial subsample seed count) was as high as 1.06 at 20 min, decreasing to 0.51 N at 60 min. The APF values of the LCT seeds were higher than those for the SCT at all seeds maturities and CTs. The average APF of the total seed readings (MB and IMG) was 0.91 and 1.49 N at 20 min CT, and they decreased to 0.37 and 0.61 N at 60 min CT for the SCT and LCT seeds, respectively (Table 3). Based on the average AFP data, it can be deduced that OCT following the cooking of the SCT seeds was below 20 min for the MB seeds, between 30 and 40 min for the IMG seeds, and between 20 and 30 min for the total seeds, while for the LCT seeds, the OCT was 40–50 min (Table 4).

The weighted average of MB (based on the count of MB intact and overcooked seeds) and IMG seeds (based on the count of IMG intact and overcooked seeds) was appreciably lower for both the SCT and LCT seeds. Therefore, based on the weighted average APF data, it can be deduced that the OCT following cooking of the SCT seeds was found to be reduced to <20 min for the MB seeds, to 20–30 min for the IMG seeds, and to <20 min for the total seeds, while for the LCT seeds, the OCT was reduced to 30–40 min for both seed maturities and the total seeds (Table 4).

However, when the weighted average of the APF of the total seeds was calculated (this included the number of intact and overcooked seeds for both seed maturities), the OCT was found to be intermediate among the two seed maturities, but closer to that of the MB seeds rather than to that of the IMG seeds. The patterns of APF value change due to cooking and seed maturity was found to be similar to those of AAPF or GAPF. The OCT of the total seeds, based on the average or weighted average of the AAPF, was 40–50 min, and that of GAPF was 30–40 min for both the SCT and LCT seeds (Table 4).

To determine the precise OCT, the distribution of the values of the texture analysis parameters were plotted against the CT (Figure 3). In each figure, the intact and overcooked MB and IMG seeds are shown. Additionally, curve fitting based on both the average and the weighted average of the MB and IMG seeds and the total seeds was performed and presented. Curve fitting followed the exponential decay model at relatively high r^2^ (>0.753) and with a statistical significance of the function of *p* > 0.05. The functions and the corresponding r^2^ and *p* values are shown in Table 4. It has been reported that the softening of legumes during cooking follows an exponential model [17].

It was observed that the use of the exponential decay model confirmed the previously estimated OCT (Table 3) from average or weighted average texture analysis parameter readings, for of all seed categories (Table 4). The advantage of using this model to determine OCT is that the OCT could be determined when it was lay either between two CTs, or even predicted when it lay outside of the experimental CT range examined. Determined or predicted CTs are also shown in Table 4: the OCT of the IMG seeds was higher in most seed samples than in the MB. The OCT of the total weighted average of the SCT was lower than that of the LCT seeds: 14.2 and 37.4 min for APF, 49.9 and 39.8-min for AAPF, and 34.1 and 40.3 min for GAPF, respectively.

Further, it was also observed that there was a deviation of the OCT produced by the model using the average compared to the weighted average texture parameter values. Such deviations were greater between the texture parameters, as well as between the seed maturities (Table 4).

By considering the determined OCT and that predicted by the model (Table 4), in view of the distribution of the values of the texture parameters during seed cooking (Figure 3), it was observed that in all CTs, an appreciable number of values exceeded the original threshold value (APF = 0.80 N, AAPF = 4.45 N.s, and GAPF = 0.10 N.s^−1^) obtained by the organoleptic test (Figure 2). This indicated that these seeds were considered as uncooked. To overcome this error, it was necessary to follow a different approach in which the uncooked seeds had to be accounted for as such.

In this work, the texture analysis parameters were expressed as the average for each seed maturity (15-seed subsample plus overcooked seeds), or for the total seeds (sum of both maturities plus overcooked seeds), as well as the weighted average in which the number of seeds within each subsample of each maturity was taken into account.

It is a common practice that OCT is estimated by the average of the set of texture analysis readings obtained in one or many CTs [6,8,13]. However, this practice may introduce an error in the estimation of the OCT, since on the one hand, this particular set of readings may not represent the whole sample and on the other, the overcooked seeds are usually excluded Figure 1 shows that the percentage of overcooked seeds at a CT of 60 min was as high as about 40% and as low 25% for the SCT and LCT, respectively [29]. Therefore, the fraction of overcooked seeds must be included in the estimation of the OCT of a particular sample.

Another common practice in OCT determination is related to sampling without considering the maturity factor within the particular sample. Thus, sampling at random may introduce an error, depending on the percentage of a maturity category within each particular sample; in Table 2 and Table 3, as well as in Figure 3, the differences in texture parameters among the two seed maturities are shown; the MB seeds were always softer than the immature seeds during CT.

### 3.5. Modeling of the Percentage of the Number of Cooked Seeds

To elaborate the second model, the distribution of seed texture parameters during CT was handled as follows. A range within each texture parameter was considered: 0–0.80 N for APF, 0–4.45 N.s for AAPF, and 0–0.1 N.s^−1^ for GAPF; the number of texture analyzer readings (corresponding to individual seeds) plus the overcooked seeds within this range for a given parameter (e.g., 0–0.80 N APF), which could be considered as cooked, was counted and expressed as a percentage of the 10 g subsample examined (within each CT, seed maturity category, and seed sample). This information is presented in Figure 4.

All values were fitted in a sigmoid curve model in which 0% accounted for hard or uncooked seeds at the beginning of the cooking procedure (0 min) and 100% for adequately cooked seeds or OCT. The model functions and the predicted values of CTD time are shown on Table 4. The predicted OCT for the range of APF readings of 0–0.80 N for the SCT and LCT seeds was 61.7 and 68.1 min, respectively, based on 99% of the seed showing APF readings within 0–0.80 N. However, the predicted OCT for the range of AAPF values of 0–4.45 N.s or of GAPF of 0–0.01 N.s^−1^ were higher than those of APF. Ranges lower or higher than 0–0.80 N AFP, 0–4.45 N.s AAPF or 0–0.1 N.s^−1^ GAPF, resulted in increased or decreased predicted CT, respectively (Table 4).

Enclosed Figure 4. A1-E1 within each graph show the correlation of the experimental vs. the predicted percentage of the cooked seeds in the ranges of 0–0.80 N APF, 0–4.45 N.s AAPF, or 0–0.1 N.s^−1^ GAPF of the total seed sample for the SCT and LCT seeds. A very strong correlation, particularly in the case of the AFP of SCT seeds, was observed. All cases showed a very strong to strong correlation, except for the AAPF/SCT seeds, which failed to correlate.

Modeling the texture analysis data (Figure 3, Table 4) aimed at determining a more precise OCT compared to that based on the average or weighted average in Table 3. Even though the OCT value in most seed of the sample X seed maturity interaction cases was double the value produced by the average or the weighted average, this was due to the fact that upon cooking at OCT, virtually all seeds (>99%) would end up cooked. Again, MB seeds exhibited a lower OCT than did the IMG, and the total seeds were intermediate between the two maturities. Scanlon et al. [17] used the value of an 80% cooked seed to determine the OCT. In general, the definition of OCT does not refer to a certain percentage of cooked seeds. More studies are needed to establish this value, both by organoleptic as well as by texture analyzer parameters, particularly APF. In any case, the sigmoid model using the percentage of cooked seeds may provide an OCT value at any percentage of cooked seed, e.g., the predicted OCT of SCT and LCT seeds for 80% cooked seeds was as low as 35.4 and 54.6 min, respectively. However, at this CT, the remaining 20% of uncooked seeds might result in a significant deviation in the organoleptic OCT and therefore, more studies are needed to establish this parameter.

Finally, it was observed that among the three texture analysis parameters, although they were all related to applied force, the APF was the only one that produced consistent and reliable OCTs. The use of AAPF resulted in much higher OCT values, and GAPF resulted in even higher values. These values were rejected along with the use of the AAPF and the GAPF parameters in determining OCT.

### 3.6. Validation of OCT Prediction Model

Table 5 shows the weighted average of the APF readings of seeds subjected to cooking at an OCT of 66.1 and 75.3 min for the SCT and LCT seeds, respectively, as was previously predicted by the sigmoid model (Table 4). Virtually all readings (except for one) were within the OCT range of 0–0.80 N, and therefore, the % of cooked seeds was 97.6% and 98.9% for the SCT and LCT seeds, respectively. The weighted average APF of the MB seeds did not differ from than that of the IMG. The weighted average of APF for total seeds was 0.64 and 0.63 N, while the tactile texture and chewiness was rated in the range of 6.8–7.3 for the SCT and LCT seeds, respectively. Validation tests further confirmed the adequate cooking based on the sigmoid model prediction of OCT.

## 4. Conclusions

This study showed that in general, lentil seed maturity categories within a seed sample differ in their 1000-seed mass, percentage of seed maturity categories, or color, as well as in mass and overcooked seed increase during CT. A difference was also observed between SCT and LCT seed samples, which reflected the differences in seed maturity categories within each seed sample. The texture analysis parameters also showed differences which reflected the seed sample origin and seed maturity differences.

Attempts to evaluate the OCT within the different seed sample x seed maturity interaction subsamples, based on average or weighted average texture analysis parameter readings during CT, resulted in non-realistic OCT values. Modeling the average or weighted average texture analysis parameter readings during CT also resulted in OCT values in which many readings were above the threshold obtained by the correlation of organoleptic with texture analysis parameter values. When the percentage of cooked seeds vs. CT was introduced to a sigmoid model, the resulted OCT obtained was realistic, but only for APF. A validation trial confirmed the predicted OCT using the sigmoid model.

Overall, the study of the seed maturities in either SCT or LCT seeds of the same cultivar revealed that the ratio of the two maturity stages/samples of origin within the given seed sample (cv Dimitra) and the related properties of each category might be very important factors for OCT prediction. This aspect could even provide the basis to more accurately evaluate the influence of genotype (lentil cultivars) and Gx E interaction on seed maturity and final seed quality characteristics. It is important and potentially beneficial for human nutrition that lentil seed maturity should be studied on a long-term basis due to the unpredictable environmental or other factors of the agricultural system involved in seed production. Therefore, to better characterize the lentil seed maturity aspect and its relation to the final seed quality, there is a need for more research on the development of seed maturation. The acknowledgement of the seed maturity factor and its interaction with genetic material is expected to enhance the understanding of seed quality

## Figures and Tables

**Figure 1 foods-12-00042-f001:**
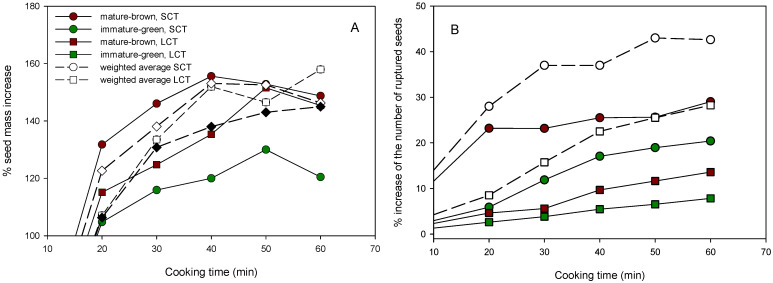
Percentage of mass increase (**A**) and increase in the number of overcooked seeds (**B**) of MB and IMG, SCT, or LCT lentil seed samples during boiling at 100 °C for 20–60 min.

**Figure 2 foods-12-00042-f002:**
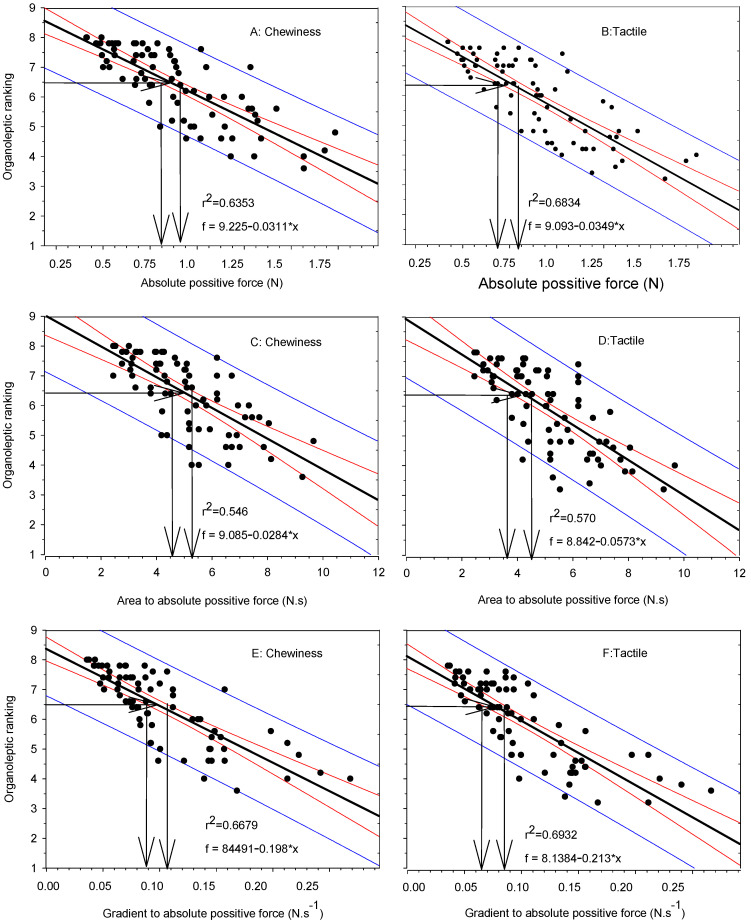
Correlation of APF (absolute positive force, (**A**,**B**)), AAPF (area to absolute positive force, (**C**,**D**)), and GAPF (gradient to absolute positive force, (**E**,**F**)) values to organoleptic chewiness and tactile firmness, respectively, of total lentil seed samples during cooking at 100 °C for 20–60 min. Black lines in bold represent the curve fitting according to linear model, red lines represent the 95% confidence intervals, and blue lines represent the prediction intervals. Arrows indicate the threshold range of the seeds, from soft to optimum cooking quality.

**Figure 3 foods-12-00042-f003:**
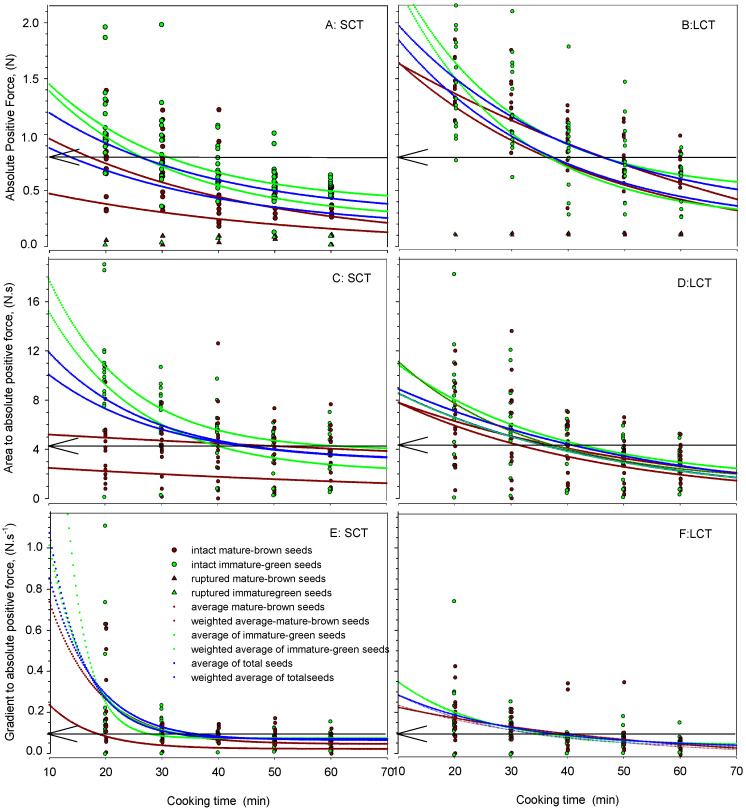
Distribution of APF (absolute positive force, (**A**,**B**)), AAPF (area to absolute positive force, (**C**,**D**)), and GAPF (gradient to absolute positive force, (**E**,**F**)) values of MB, IMG, and total lentil seed samples, of either easy (SCT) or hard (LCT) to cook lentil seeds, under cooking at 100 °C for 20–60 min. Each sample consisted of ~300 seeds per cooking time, of which 30 seeds (15 MB and 15 IMG) were subjected to texture analysis. Solid lines represent curve fitting according to the exponential decay model. The arrow indicates the threshold of optimum cooking obtained from Figure 2.

**Figure 4 foods-12-00042-f004:**
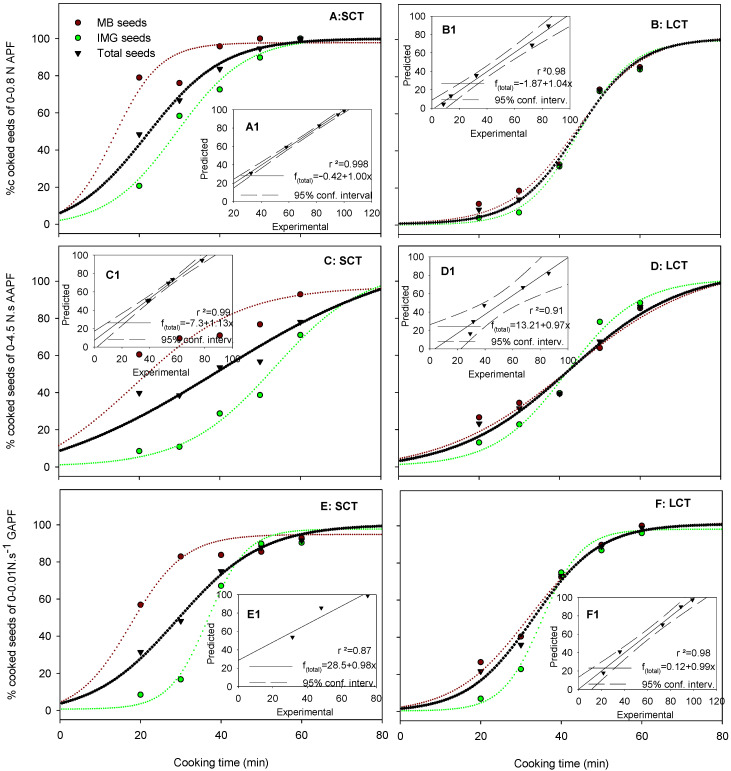
The evolution of the percentage of cooked seeds based on APF (absolute positive force, (**A**,**B**)), AAPF (area to absolute positive force, (**C**,**D**)), and GAPF (gradient to absolute positive force, (**E**,**F**)) values of MB, IMG, and total lentil seed samples, of either easy (SCT) or hard (LCT) to cook lentil seeds, during cooking at 100 °C for 20–60 min. Dashed lines represent curve fitting according to the sigmoid model. The enclosed figure within each graph shows the correlation of experimental versus predicted values of the total seed sample.

**Table 1 foods-12-00042-t001:** Visual characteristics and mass of 1000 lentil seeds based on the seed maturity (MB and IMG seeds) of SCT and LCT lentil seeds.

Seed Sample	1000-Seed Mass, Whole Sample (g)	1000-Seed Mass, MB Seeds (g)	1000-Seed Mass, IMG Seeds (g)	ΜΒ Seeds (%)	IMG Seeds (%)	Whole Sample Lightness (L*)	Whole Sample Redness (a*)	Whole Sample Yellowness (b*)
SCT	24.4 B	33.35 A	16.65 B	47.6 B	52.4 A	56.0 B	3.2 B	25.2 A
LCT	34.1 A	33.64 A	35.49 A	75.0 A	25.0 B	61.6 A	5.1 A	25.9 A

Different letters following values within each factor and column indicate significantly different values at the 0.05 level, according to Duncan’s multiple range test.

**Table 2 foods-12-00042-t002:** Analysis of variance for average texture analysis parameters APF (absolute positive force), AAPF (area to absolute positive force), and GAPF (gradient to absolute positive force) of short (SCT) and long (LCT) cooking time lentil seeds, respectively, during cooking at 100 °C for 20–60 min.

Source of Variability	DF	APF (g)		AAPF (g∙s)		GAPF (g∙s^−1^)	
		η^2^	*P*	η^2^	*P*	η^2^	*P*
Seed sample (A)	1	26.96	***	25.07	***	26.28	***
Cooking time (B)	4	61.24	***	58.55	***	63.18	***
Seed maturity (C)	1	3.80	***	6.99	***	0.53	NS
(A) × (B)	4	5.53	**	4.50	*	8.22	**
(A) × (C)	1	0.26	NS	0.28	NS	0.07	NS
(B) × (C)	4	1.30	NS	6.87	NS	2.33	NS
(A) × (B) × (C)	4	0.15	NS	0.58	NS	0.50	NS
		Means					
Seed sample (A)	SCT	0.68 B		3.91 B		0.06 B	
	LCT	1.03 A		5.75 A		0.12 A	
Cooking time (B)	20 min	1.27 A		6.84 A		0.16	
	30 min	1.04 B		6.10 A		0.11 B	
	40 min	0.76 C		4.34		0.08 C	
	50 min	0.64 D		3.58 BC		0.06 CD	
	60 min	0.55 D		3.28 C		0.05 D	
Seed maturity (C)	MB	0.79 B		4.34 B		0.09 A	
	IMG	0.92 A		5.31 A		0.09 A	

DF: degrees of freedom; η^2^: eta squared; *P*: probability; *, significant at 0.05 level; **, significant at 0.01 level; ***, significant at 0.001 level; NS non-significant. Different letters following values within each factor and column indicate significantly different values at the 0.05 level, according to Duncan’s multiple range test.

**Table 3 foods-12-00042-t003:** Average and weighted APF (absolute positive force), AAPF (area to absolute positive force), and GAPF (gradient to absolute positive force) values of MB, IMG, and total lentil seed samples, either easy (SCT) or hard (LCT), during cooking at 100 °C for 20–60 min.

Cooking Time (min)	AFP (g)				AAFP (g∙s)				GAFP (g∙s^−1^)			
	SCT	LCT	SCT	LCT	SCT	LCT	SCT	LCT	SCT	LCT	SCT	LCT
	Average		Weighted Average		Average		Weighted Average		Average		Weighted Average	
	Mature-brown											
20	0.76 ± 0.35 A	1.34 ± 0.53 A	0.37 ± 0.17	1.23 ± 0.48	4.94 ± 3.06 A	6.71 ± 3.56 A	2.40 ± 1.48	6.13 ± 3.25	0.25 ± 0.22 A	0.19 ± 0.11 A	0.09 ± 0.08	0.17 ± 0.10
30	0.73 ± 0.32 A	1.18 ± 0.48 A	0.34 ± 0.15	0.99 ± 0.40	4.55 ± 2.14 A	6.45 ± 3.93 A	2.11 ± 0.99	5.45 ± 3.32	0.09 ± 0.03 B	0.13 ± 0.05 B	0.05 ± 0.02	0.11 ± 0.05
40	0.46 ± 0.30 B	0.87 ± 0.37 B	0.23 ± 0.15	0.67 ± 0.29	4.47 ± 3.25 B	4.50 ± 2.30 B	2.29 ± 1.66	3.46 ± 1.77	0.09 ± 0.03 BC	0.11 ± 0.09 B	0.03 ± 0.01	0.09 ± 0.07
50	0.45 ± 0.18 B	0.73 ± 0.33 B	0.22 ± 0.09	0.55 ± 0.29	4.00 ± 2.34 B	3.56 ± 2.06 B	2.00 ± 1.17	2.69 ± 1.56	0.08 ± 0.04 CD	0.08 ± 0.08 BC	0.04 ± 0.02	0.06 ± 0.06
60	0.38 ± 0.14 B	0.57 ± 0.21 B	0.14 ± 0.05	0.42 ± 0.16	3.51 ± 2.18 B	2.86 ± 1.61 B	1.30 ± 0.81	2.12 ± 1.20	0.06 ± 0.03 D	0.05 ± 0.02 C	0.02 ± 0.01	0.04 ± 0.02
	Immature green											
20	1.06 ± 0.46 A	1.63 ± 0.53 A	0.97 ± 0.42	1.50 ± 0.75	10.78 ± 3.06 A	8.94 ± 3.56 A	9.87 ± 4.34	8.22 ± 4.28	0.29 ± 0.22 A	0.21 ± 0.11 A	0.27 ± 0.27	0.19 ± 0.15
30	0.83 ± 0.43 B	1.18 ± 0.48 B	0.75 ± 0.39	1.00 ± 0.38	4.55 ± 2.14 AB	6.45 ± 3.93B	6.58 ± 2.37	5.76 ± 2.89	0.09 ± 0.03 B	0.13 ± 0.05 B	0.12 ± 0.05	0.12 ± 0.05
40	0.67 ± 0.25 BC	0.87 ± 0.37 C	0.53 ± 0.20	0.71 ± 0.25	4.47 ± 3.25 BC	4.50 ± 2.30 BC	4.62 ± 1.71	4.23 ± 1.71	0.09 ± 0.03 BC	0.11 ± 0.09 C	0.07 ± 0.02	0.07 ± 0.02
50	0.54 ± 0.24 C	0.73 ± 0.26 C	0.41 ± 0.18	0.52 ± 0.18	4.49 ± 1.77 C	3.83 ± 1.69 C	3.42 ± 1.35	2.71 ± 1.19	0.08 ± 0.03 C	0.08 ± 0.04 C	0.06 ± 0.02	0.05 ± 0.03
60	0.51 ± 0.12 C	0.65 ± 0.19 C	0.38 ± 0.12	0.40 ± 0.12	4.24 ± 1.86 C	3.58 ± 1.37 C	3.17 ± 1.39	2.21 ± 0.84	0.07 ± 0.03 C	0.06 ± 0.03 C	0.05 ± 0.02	0.04 ± 0.02
	total seeds											
20	0.91 ± 0.43 A	1.49 ± 0.69 A	0.73 ± 0.31	1.32 ± 0.57	7.86 ± 4.92 A	7.83 ± 4.23 A	8.69 ± 3.06	7.47 ± 3.58	0.27 ± 0.26 A	0.20 ± 0.14 A	0.28 ± 0.18	0.19 ± 0.12
30	0.78 ± 0.38 B	1.19 ± 0.46 B	0.60 ± 0.29	1.01 ± 0.40	5.88 ± 2.74 A	6.67 ± 3.66 A	6.35 ± 1.78	6.58 ± 3.22	0.11 ± 0.05 B	0.13 ± 0.06 B	0.12 ± 0.04	0.13 ± 0.05
40	0.56 ± 0.29 C	0.88 ± 0.34 C	0.41 ± 0.18	0.70 ± 0.28	5.09 ± 2.82 BC	4.90 ± 2.23 B	5.21 ± 1.69	4.77 ± 1.75	0.09 ± 0.03 BC	0.10 ± 0.07C	0.09 ± 0.02	0.12 ± 0.06
50	0.41 ± 0.17 C	0.73 ± 0.29 CD	0.49 ± 0.21	0.57 ± 0.23	4.24 ± 2.07 BC	3.69 ± 1.87 BC	4.28 ± 1.26	3.64 ± 1.48	0.08 ± 0.03 CD	0.08 ± 0.06 CD	0.08 ± 0.02	0.11 ± 0.05
60	0.37 ± 0.10 C	0.61 ± 0.20 D	0.45 ± 0.14	0.45 ± 0.15	3.86 ± 2.04 C	3.20 ± 1.52 C	3.96 ± 1.13	3.07 ± 1.13	0.07 ± 0.03 D	0.06 ± 0.00 D	0.07 ± 0.02	0.10 ± 0.02

Different letters following values within each factor and column indicate significantly different values at the 0.05 level, according to Duncan’s multiple range test.

**Table 4 foods-12-00042-t004:** Estimated range and predicted OCT based on the average and weighted average using the exponential decay model or based on the percentage of cooked seeds using the sigmoid model of MB, IMG, or total lentil seed samples (easy to cook—SCT, and hard to cook—LCT) for APF (absolute positive force), AAPF (area to absolute positive force), and GAPF (gradient to absolute positive force) during cooking at 100 °C for 20–60 min.

Seed Sample	Seed Maturity	Model 1: Exponential Decay	r^2^	*p* Value	Estimated OCT Time (min)	Model 1 Predicted OCT (min)	Model 1: Exponential Decay	r^2^	*p* Value	Estimated OCT Time (min)	Model 1 Predicted OCT (min)	Model 2: Sigmoid. 3rd Parameter	r^2^	*p* Value	Model 2 Predicted OCT (min)	% Cooked Seeds
		Average					Weighted Average					% Cooked Seeds				
		APF														
SCT	MB	f = 9.86 + 112.2 * exp(−0.023 * x)	0.8942	ns	<20	<10	f = 2.51 + 60.23 * exp(−0.021 * x)	0.9401	**	<20	<10	f_(mat.−brown)_ = 100.1/(1 − exp(−(x − 13.88)/5.6))	0.9964	***	42	97
	IMG	f = 39.24 + 170.1 * exp(−0.044 * x)	0.9960	**	30–40	31.5	f = 23.34 + 182.14 * exp(−0.043 * x)	0.992	**	20–30	26.8	f_(imm.−green)_ = 101.79/(1 − exp(−(x − 29.28)/8.6))	0.9960	***	67.7	99
	Total	f = 27.21 + 133.8 * exp(−0.034 * x)	0.9782	*	20–30	26.8	f = 14.79 + 102.9 * exp(−0.031 * x)	0.934	ns	<20	14.2	F_(total)_ = 103.5/(1 − exp(−(x − 21.9)/9.4))	0.9964	***	66.1	99
LCT	MB	f = −71.02 + 268.1 * exp(−0.012 * x)	0.9868	*	40–50	46.3	f = 1.14 + 218.5 * exp(−0.027 * x)	0.9870	*	30–40	36.4	f_(mat.−brown)_ = 101.16/(1 − exp(−(x − 31.1)/8.4))	0.9948	***	74.29	99
	IMG	f = 48.71 + 319.4 * exp(−0.049 * x)	0.9988	**	40–50	46.3	f = 21.88 + 343.9 * exp(−0.048 * x)	1	***	40–50	36.2	f_(imm.−green)_ = 99.62/(1 − exp(−(x − 44.83)/6.5))	0.9968	***	79.21	99
	Total	f = 26.41 + 240.18 * exp(−0.032 * x)	0.9959	**	40–50	46.3	f = 16.73 + 243.9 * exp(−0.035 * x)	0.995	**	30–40	37.4	F_(total)_ = 100.27/(1 − exp(−(x − 44.5)/7.5))	0.9964	***	75.3	99
		AAPF														
SCT	MB	f = 600.3 * exp(−0.0048 * x)	0.7533	ns	40–50	59.45	f = 312.7 * exp(−0.0107 * x)	0.671	ns	<20	<10	f_(mat.−brown)_ = 106.09/(1 − exp(−(x − 18.88)/13.22))	0.9702	***	86.32	99
	IMG	f = 430.9 + 3020.4 * exp(−0.0704 * x)	0.9991	***	40–60	>70	f = 240.7 + 2557.7 * exp(−0.0601 * x)	0.938	**	40–50	41.6	f_(imm.−green)_ = 88.03/(1 − exp(−(x − 12.99)/4.75))	0.9951	***	81.95	99
	Total	f = 331.1 + 1245.5 * exp(−0.0490 * x)	0.9952	**	40–50	47.7	f = 343.4 + 1656.5 * exp(−0.0559 * x)	0.999	**	30–40	48.9	F_(total)_ = 127.83/(1 − exp(−(x − 40.19)/23.9))	0.9767	***	85.78	99
LCT	MB	f = 8.01 + 4169.6 * exp(−0.2730 * x)	0.9856	*	20–30	28.0	f = 2.77 + 76.55 * exp(−0.1253 * x)	0.961	*	<20	18.9	f_(mat.−brown)_ = 101.16/(1 − exp(−(x − 31.1)/8.4))	0.9948	***	81.40	99
	IMG	f = 7.12 + 283.2 * exp(−0.1258 * x)	0.9995	***	20–30	36.7	f = 5.11 + 283.1 * exp(−0.1150 * x)	1	***	30–40	33.4	f_(imm.−green)_ = 99.62/(1 − exp(−(x − 44.83)/6.5))	0.9968	***	75.93	99
	Total	f = 7.61 + 517.2 * exp(−0.1620 * x)	0.9966	**	30–40	33.2	f = 7.51 + 441.5 * exp(−0.1523 * x)	0.998	**	30–40	34.1	F_(total)_ = 100.27/(1 − exp(−(x − 44.57)/7.5))	0.9964	***	80.31	99
		GAPF														
SCT	MB	f = 1100.4 * exp(−0.0211 * x)	0.9369	**	40–50	42.3	f = 1162.1 * exp(−0.0268 * x)	0.9640	**	30–40	35.5	f_(mat.−brown)_ = 96.87/(1 − exp(−(x − 17.59)/6.6))	0.99	***	>100	97
	IMG	f = 164.9 + 1528.5 * exp(−0.0353 * x)	0.9916	**	40–50	47.3	f = 150.7 + 1695.1 * exp(−0.0421 * x)	0.992	**	40–50	41.1	f_(imm.−green)_ = 97.05/(1 − exp(−(x − 36.76)/4.8))	0.9960	***	>100	97
	Total	f = −59.7 + 1325.1 * exp(−0.0210 * x)	0.9831	*	40–50	45.6	f = −2.91 + 1262.7 * exp(−0.0257 * x)	0.986	*	40–50	39.8	F_(total)_ = 100.6/(1 − exp(−(x − 21.0)/6.7))	0.9975	***	61	99
LCT	MB	f = −4.44 + 34.8 * exp(−0.0208 * x)	0.9700	*	40–50	42.5	f = −0.64 + 35.1 * exp(−0.0329 * x)	0.9752	*	30–40	36.7	f_(mat.−brown)_ = 101.16/(1 − exp(−(x − 31.1)/8.4))	0.9948	***	65.00	99
	IMG	f = 5.01 + 60.0 * exp(−0.0646 * x)	0.9919	**	40–50	38.50	f = 4.07 + 64.7 * exp(−0.0681 * x)	0.992	**	30–40	35.31	f_(imm.−green)_ = 99.62/(1 − exp(−(x − 44.83)/6.5))	0.9968	***	>100	97
	Total	f = 3.08 + 45.1 * exp(−0.052 * x)	0.9948	**	40–50	36	f = 3.46 + 42.1 * exp(−0.0465 * x)	0.996	**	20–30	40.2	F_(total)_ = 101.37/(1 − exp(−(x − 33.1)/8.2))	0.9976	***	65.54	99

*, significant at 0.05 level; **, significant at 0.01 level; ***, significant at 0.001 level; ns, non-significant.

**Table 5 foods-12-00042-t005:** Weighted average of APF (absolute positive force) tactile texture and chewiness ± standard deviation, as well as percentage of cooked seeds of easy-to-cook SCT and hard-to-cook LCT seed samples during cooking at 100 °C at previously predicted optimal cooking time (OCT).

Seed Sample	Predicted OCT (min)	Applied CT (min)	APF (N)	Tactile	Chewiness	Cooked Seeds (%)
SCT	66.1	66.1	0.64 ± 0.13	6.8 ± 0.94	7.1 ± 2.1	97.6
LCT	75.3	75.3	0.63 ± 0.11	7.0 ± 0.60	7.3 ± 6.0	98.9

## Data Availability

The data are avaliable from the corresponding author. The data are not publicly available due to privacy restrictions.

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
