# Peer review of "Modeling the Effects of Seed Maturity on Cooking Time of ‘Dimitra’ Lentils"

_foods, 2022, doi:10.3390/foods12010042_

Round 1

Reviewer 1 Report

I have reviewed the manuscript titled: Modeling the effects of seed maturity on cooking time of “Dimitra” lentils. This article aims to study the seed maturity effects on optimum cooking time of lentil seeds “cv ‘Dimitra’ of short cooking time (brown color) and long cooking time (green color) of mature and immature seed, respectively” using a model. The information of this work is very useful and relevant and there are several physical properties such as absolute positive force, area to absolute positive force, gradient to absolute positive force were monitored during cooking of 20-60 min by correlating of organoleptic. The article is innovative and interesting for lentil processing manufacture, it contains original and interesting information to understand the correlation among texture analysis and optimum cooking time with the organoleptic.

The abstract was well addressed including 1000 seed mass and percentages of maturity categories in lentils samples and optimal cooking time thresholds. The information of modeling of the percentage of cooked seeds during cooking time, predicted a realistic optimal cooking time were introduced. And the effect of maturity categories within short cooking and long cooking time seeds was also mentioned. This article also found that texture analysis parameters averaging and weighted averaging would not to produce realistic optimal cooking.

The introduction referred lentil consumption and its cooking qualities relating to genetic factor, chemical composition and physicochemical characteristic like size, maturity and processing conditions. All the above factors will relate to optimal cooking time. The sensory and instrument methods for determined optimal cooking time were also mentioned.

Materials and Methods section well address lentil seed color and moisture content measurement. The assessment of cooking quality including cooking procedure, texture analysis, establishing optimal cooking time, model development and validation for optimal cooking prediction model.

The Results and Discussion section were well written. This study successfully establishes organoleptic value to adopt to adequate cooking doneness of lentil seeds and correlating to both tactile texture and chewiness. Although, texture analysis parameters averaging and weighted averaging and their modeling can not produce realistic optimal cooking time because texture values exceeding optimal cooking time threshold. However,modeling of the percentage of cooked seeds during cooking time, predicted a realistic optimal cooking time and it was validated. This study demonstrated a differences between sort cooking time and long cooking seed samples reflecting the differences of seed maturity groups within each sample. Validation test of optimal cooking time predict an adequate cooking based on sigmoid model.

I am not a native English speaker. The references are following the foods’ reference format except the specie name of bean and lentil should be italic. The journal name of Reference #17 “CCHEM” should revise to “Cereal Chem”. All the revised suggestions were as attached file in yellow marked.

I enjoyed reading this manuscript; the needs of special groups of lentil and bean processing manufacturer and researcher. This manuscript presents some interesting data and useful statistic analysis information for predicting optimal cooking time and sensorial evaluation.

Author Response

Thank you for your commends.

All  references were revised following the foods’ reference format including the specie name of bean and lentil in italic.

The journal name of Reference #17 “CCHEM” was revised to “Cereal Chem”.

Other marks in yellow were corrected.

Reviewer 2 Report

The study is quite interesting. However, the English style and many abbreviated words made it difficult to follow some aspects of the paper.

Minor comments:

Line 125:Delete “was”.

Line 136: "2.3. Assessment of cooking quality". There should be a section on experimetal design where the variables are clearly defined. The way the samples were explained in section 2.3.1 may confuse the readers.

Line 467 and 468: "x" should be "and".

Author Response

Thank you for your review.

Errrors in lines 125, 467 and 468:were corrected.

In line 136: it was suggested that a section on experimetal design where the variables are clearly defined should be incorporated.

Although, this is explained in statistical analysis, a brief section was also included in:

2.1. Plant Materials and Experimental Design

Seed...cooking quality study (was) started. 

The experimental design apart of the  two seed samples (SCT and LCT) included two seed maturities per seed sample (brown and green seeds) in addition to the cooking time factor (20, 30, 40, 50, and 60 min) for all seeds.